Spatial patterns of coral survivorship: impacts of adult proximity versus other drivers of localized mortality

Gibbs David A. 1 3
Hay Mark E. 2 mark.hay@biology.gatech.edu
1 School of Biology, Georgia Institute of Technology , Atlanta, GA , United States
2 School of Biology and Aquatic Chemical Ecology Center, Georgia Institute of Technology , Atlanta, GA , United States
3 Current affiliation: Tetra Tech, Inc. , Atlanta, GA , United States
Kramer Donald
Electronic publication date: 2015 Nov 24
Publication date: 2015
Volume: 3
Electronic Location ID: e1440
Received 2015 Aug 8; Accepted 2015 Nov 4
Copyright: © 2015 Gibbs and Hay
Copyright year: 2015
Copyright holder: Gibbs and Hay
License: This is an open access article distributed under the terms of the Creative Commons Attribution License, which permits unrestricted use, distribution, reproduction and adaptation in any medium and for any purpose provided that it is properly attributed. For attribution, the original author(s), title, publication source (PeerJ) and either DOI or URL of the article must be cited.
License URL: https://creativecommons.org/licenses/by/4.0/

Keywords: Pocillopora, Seriatopora, Janzen–Connell hypothesis, Balistapus, Predation

Funding: National Science Foundation OCE 0929119 National Institutes of Health U01-TW007401 U19TW007401 Georgia Institute of Technology Funding came from the National Science Foundation (OCE 0929119), the National Institutes of Health (U01-TW007401 and U19TW007401), and the Teasley Endowment to the Georgia Institute of Technology. The funders had no role in study design, data collection and analysis, decision to publish, or preparation of the manuscript.

==============================
Species-specific enemies may promote prey coexistence through negative distance- and density-dependent survival of juveniles near conspecific adults. We tested this mechanism by transplanting juvenile-sized fragments of the brooding corals Pocillopora damicornis and Seriatopora hystrix 3, 12, 24 and 182 cm up- and down-current of conspecific adults and monitoring their survival and condition over time. We also characterized the spatial distribution of P. damicornis and S. hystrix within replicate plots on three Fijian reef flats and measured the distribution of small colonies within 2 m of larger colonies of each species. Juvenile-sized transplants exhibited no differences in survivorship as a function of distance from adult P. damicornis or S. hystrix. Additionally, both P. damicornis and S. hystrix were aggregated rather than overdispersed on natural reefs. However, a pattern of juveniles being aggregated near adults while larger (and probably older) colonies were not suggests that greater mortality near large adults could occur over longer periods of time or that size-dependent mortality was occurring. While we found minimal evidence of greater mortality of small colonies near adult conspecifics in our transplant experiments, we did document hot-spots of species-specific corallivory. We detected spatially localized and temporally persistent predation on P. damicornis by the territorial triggerfish Balistapus undulatus. This patchy predation did not occur for S. hystrix. This variable selective regime in an otherwise more uniform environment could be one mechanism maintaining diversity of corals on Indo-Pacific reefs.

Introduction

The processes maintaining high numbers of species in tropical rainforests and coral reefs have long been investigated (Connell, 1978). One suggested mechanism for maintaining diversity is the Janzen–Connell hypothesis (Janzen, 1970; Connell, 1971), which proposes that species-specific enemies clustered near adults increase the local mortality of conspecific juveniles and prevent any single species from monopolizing resources. It has generally been applied to long-lived, stationary, terrestrial organisms such as trees (Zhu et al., 2013). Although there are examples of species-specific distance- or density-dependent mortality affecting community species richness (e.g., Packer & Clay, 2000; Petermann et al., 2008; Bagchi et al., 2014), a meta-analysis found no general, net effect of distance from parent on offspring mortality across a variety of plant types, habitats, or life stages (Hyatt et al., 2003). Thus, some tree species may experience Janzen–Connell effects (Johnson et al., 2012) but the generality of the pattern has been difficult to document (Hyatt et al., 2003). In part, this may be because numerous other processes (habitat heterogeneity, spatial patterns of competitors, etc.) could obscure Janzen–Connell effects. This makes experimental tests difficult in field settings, especially when spatial scales over which they may be relevant are unclear.

Research addressing Janzen–Connell effects on coral reefs is rare (Marhaver et al., 2013). Explanations for maintenance of coral diversity more often invoke disturbance regimes, abiotic gradients (e.g., light, sedimentation), and competition hierarchies (Lang, 1973; Connell, 1978; Buss & Jackson, 1979; Porter et al., 1981). One reason for the paucity of tests in marine systems may be that the hypothesis assumes that dispersal decreases monotonically with distance from parents and that the average dispersal distance is greater than the average distance from the parent at which predation occurs but that they are on the same order of magnitude (Nathan & Casagrandi, 2004), neither of which necessarily applies to marine species with pelagic larvae. Coral larvae are competent to settle within hours of release to months after release (Richmond, 1987; Miller & Mundy, 2003; Nozawa & Harrison, 2008) and may disperse up to hundreds of kilometers (Jones et al., 2009; Torda et al., 2013). Therefore, unlike seeds of many tree species, coral larvae need not be distributed as “seed shadows” with juveniles clustered near parents.

Nevertheless, distance- or density-dependent mortality of juveniles could affect coral species employing either of the sexual reproduction methods that corals use: broadcast spawning, in which eggs and sperm are released into the water column and fertilized eggs develop outside corals, or brooding, in which sperm are released into the water column and fertilize eggs retained inside adult coral colonies. Larvae produced by both methods select their settlement sites and can be attracted to the chemical cues of conspecifics (Dixson, Abrego & Hay, 2014); this could lead to larvae settling near conspecific adults or in aggregations (e.g., Dunstan & Johnson, 1998). Additionally, brooding corals may cast larval shadows akin to terrestrial seed shadows because larvae from brooding corals frequently settle quickly and close to their parents (Carlon & Olson, 1993; Tioho, Tokeshi & Nojima, 2001; Vermeij, 2005; Vermeij & Sandin, 2008; Torda et al., 2013).

Selective mortality of juveniles near conspecific adults is often assumed to be due to specialist enemies that accumulate near adults over their lifetimes. While this may be the case for terrestrial plants where many herbivores and pests are specialists (Bernays, 1989), it is unclear to what extent this applies to corals, of which there are relatively few identified species-specific consumers that might be expected to accumulate near adults of specific prey species (Cornell & Karlson, 2000; Rotjan & Lewis, 2008; but see Neudecker, 1979 and Jayewardene, Donahue & Birkeland, 2009 for examples of coral-specific predation).

However, there is growing evidence from both terrestrial (Packer & Clay, 2000; Bagchi et al., 2014; Fricke, Tewksbury & Rogers, 2014) and marine systems (Marhaver et al., 2013) that microbial pathogens may accumulate near adults and suppress the survivorship of conspecific recruits or juveniles. In the most direct test of the Janzen–Connell hypothesis in corals, Marhaver et al. (2013) used a series of lab and field investigations in the Caribbean to attribute higher mortality of Orbicella (formerly Montastraea) faveolata recruits placed near adult conspecifics to adult-associated microbial enemies. They found a complex relationship between distance from adult colonies, current direction, and recruit mortality. In less direct tests, Vermeij (2005) and Vermeij & Sandin (2008) observed that survival of coral recruits decreased with increasing cover of conspecifics; they hypothesized that this was due to species-specific microorganisms rather than to saturation of a limiting resource.

No study has explicitly tested the Janzen–Connell hypothesis in brooding corals or in fragments typical of juvenile-sized corals. These conditions may generate different results from previously tested conditions because brooding corals may be subject to different distance-dependent mortality patterns compared to broadcast spawning corals and because small fragments may differ from newly settled larvae. For instance, distance-dependent mortality is known to affect some seedlings more than seeds in terrestrial systems (Hyatt et al., 2003). We experimentally evaluated distance-dependent mortality of juvenile-sized corals in the field and correlatively surveyed multiple reefs for patterns of spatial distribution suggestive of Janzen–Connell effects. We focused on two brooding coral species (Seriatopora hystrix and Pocillopora damicornis) whose planulae recruit over short distances, the latter of which is known to be a preferred prey for some coral consumers (Neudecker, 1979).

Methods

Study site characteristics

This study was conducted on reef flats within no-take marine protected areas (MPAs) adjacent to Votua, Vatu-o-lailai, and Namada villages along the Coral Coast of Viti Levu, Fiji. These reserves are scattered along 11 km of fringing reef and are separated by ∼3–8 km. The reserves have high coral cover (38–56%), low macroalgal cover (1–3%), and a high biomass and diversity of herbivorous fishes (Rasher, Hoey & Hay, 2013; Bonaldo & Hay, 2014). The reef flats range from ∼1–3 m deep at high tide, exposed to ∼1.5 m deep at low tide, extend ∼500–600 m from shore to the reef crest, and are typical of exposed reef flats occurring throughout Fjii.

Except during low tides in calm weather, waves push water over the reef front, and water then flows laterally across the reef flats to discharge through channels bisecting the flats. This creates a relatively predictable current direction at most locations.

Survival experiments

To test whether juvenile corals experienced distance-dependent mortality near adult conspecifics, we created ∼5 mm tall fragments of P. damicornis and S. hystrix, selected suitable adult focal colonies (defined below), attached conspecific fragments 3, 12, 24 and 182 cm up- and down-current from each focal adult, and monitored fragment survival. General direction of current (east to west) was determined by in-water observations over ∼7 years of working at this site. We conducted this experiment in Votua village’s MPA, which supports a diverse assemblage of corals covering about 50% of hard substrates (Rasher, Hoey & Hay, 2013). Water flow across this reef flat is often negligible around low tide, reducing the potential for constantly dissipating microbial enemies away from focal colonies.

We used pliers to clip 16 fragments of 30–40 polyps each from the tips of each of 24 large P. damicornis and 24 large S. hystrix colonies in the Votua village MPA. We employed fragments from older colonies as proxies for ∼6 month old juveniles (Sato, 1985) because, despite these species reproducing monthly in some locations (Fan et al., 2002; Kuanui et al., 2008), neither species planulated at our site during the months of this study (August through October 2013). The fragments from each of four source colonies for a species were collected in six rounds over two days. Each round was taken to shore and four fragments (one from each source colony) were epoxied (Emerkit epoxy) onto the unglazed side of 16 2.54 × 2.54 cm tiles. Thus, each tile had fragments from four different colonies and sets of 16 tiles had fragments from the same four colonies of the same species. After epoxying, tiles were held in a tub of seawater for ∼1 h, allowing the epoxy to harden. Tiles were then cable-tied onto metal racks at ∼1 m deep in the MPA and allowed to acclimate for two weeks before deployment in the experiment. Survivorship during acclimation was 100%, producing 384 fragments on 96 tiles for each coral species.

Within the MPA, 10 adult P. damicornis and 10 adult S. hystrix colonies served as focal colonies. Focal colonies: (i) were >10 cm at their smallest diameter (10 to 35 cm for P. damicornis and 10 to 75 cm for S. hystrix), (ii) had no conspecific colonies within 4 m (so as not to confound effects of the focal colony with effects of nearby conspecifics), (iii) were 5–40 cm deep at low tide, and (iv) had space for 190 cm PVC pipes to be placed roughly east and west (the predominant current direction was west) without disturbing other corals. Focal colonies were photographed from above and their size determined using ImageJ (Rasband, 1997).

Twenty mm diameter by 190 cm long PVC pipes served as platforms to which we attached the tiles. Pipes were anchored to the reef by driving steel rebar through pre-drilled holes and cementing the rebar to the pipe. Notches 2.54 cm long allowed us to cable-tie tiles onto the pipes at distances of 3, 12, 24 and 182 cm from focal colonies (Figs. 1A and 1B). This approach secured all pipes and tiles throughout the experiment. These distances and this scale were chosen to match a previous experiment in the Caribbean that had detected distance-dependent mortality of newly settled recruits for a broadcast spawning coral (Marhaver et al., 2013).

Figure 1 Seriatopora hystrix fragments on tiles on PVC array.

Experimental set up of coral fragments epoxied to tiles on PVC pipes around focal adult colonies. (A) PVC pipes extending upstream and downstream from a focal Seriatopora hystrix colony. (B) Fragments epoxied onto ceramic tiles around a S. hystrix colony.

Tiles were randomly assigned to positions on pipes. Thus, fragments at each distance and around each conspecific focal colony were random with respect to source colony. Unassigned tiles were kept on the rack as spares (64 fragments on 16 tiles for each coral species).

Every 1–2 d after deployment, we examined all fragments, recording survivorship, consumption, overgrowth by algae, bleaching, or other changes in status. The P. damicornis fragments were observed for 59 days. The S. hystrix fragments were deployed one month later and observed for 29 days.

On some P. damicornis tiles, three or four of the fragments disappeared within a 24 h period between checks on their condition, appearing to have been bitten off. To determine the agents of this localized mortality, we replaced tiles whose four fragments had been eaten with spare tiles holding four healthy fragments around three of the focal colonies that had experienced localized mortality and videotaped the tiles (GoPro II HD) from about 1 m away during the following high tides. Cameras were retrieved at the next low tide and the videos watched.

We evaluated overall survival patterns and mortality specifically due to bleaching or predation using mixed-effects Cox proportional hazards survival models (coxme package, Therneau, 2012) in R (R Core Team, 2014). Distance and direction from focal colony were fixed effects (4 levels and 2 levels, respectively) and focal colony and tile nested within focal colony were random effects because fragments were blocked by tile and focal colony. The size of the focal colony and the depth of the tiles were included as covariates. Finally, we compared the relative levels of predation and bleaching in P. damicornis and S. hystrix using a chi-square test.

Distribution surveys

We characterized the spatial distribution of P. damicornis and S. hystrix in the reef flat MPAs of Namada, Vatu-o-lailai, and Votua villages at two scales (August through October 2013). For the larger-scale survey, we mapped each colony within 8 × 8 m plots (N = 5, 5, and 10 for Namada, Vatu-o-lailai, and Votua, respectively). Each plot was divided into 256 0.5 × 0.5 m cells and each coral ≥1 cm across mapped into a cell. The location of each survey plot was determined by randomly choosing a point on shore, swimming 100, 200, or 300 kicks directly away from shore at that point, and surveying the closest bommie (coral patch) large enough to fill more than three quarters of an 8 × 8 m plot. In four of 10 surveys at Votua and in all five surveys at Vatu-o-lailai and Namada, we also measured the largest diameter of each P. damicornis colony. We did not measure S. hystrix colony size because the frequent discontinuity of colonies made accurate estimation of colony area too error-prone. To avoid confounding biotically-driven spatial distribution with patterns caused by patchiness of suitable substrate, we also recorded which cells were comprised primarily of unsuitable habitats such as sand-scoured pools or channels and bommie tops covered in rubble.

We analyzed these data using the neighborhood density function O(r) in the point pattern analysis program Programita (Wiegand & Moloney, 2004). This analysis identifies distances at which individuals are aggregated, randomly spaced, or overdispersed compared to a specified null model. Unlike the more frequently used Ripley’s K(r) statistic, each distance category is not affected by those inside it; expected aggregation at each distance is compared to the observed value independently of nearer distances. Each concentric ring centered on an individual coral is separately placed on the aggregated-overdispersed continuum and displays the spatial pattern within a different distance category. Ring width was 0.5 m extending up to 4 m. The null model for this analysis was complete spatial randomness (CSR). Because the variance in substrate types violated CSR’s assumption of uniform likelihood of coral placement, we conducted the below analyses once using the entirety of all 8 × 8 m plots and a second time excluding cells of unsuitable habitat (which should better meet CSR’s assumption of uniform likelihood).

To determine whether the observed spatial pattern was random, significantly aggregated, or overdispersed, Programita simulated placement of each plot’s colonies 999 times using CSR, calculated O(r) for each simulation, then combined replicate O(r)’s from each reef and from all three reefs. This generated a distribution of simulated O(r)’s from which we established the significance of the observed spatial patterns. The distance(s) at which significant aggregation or overdispersion occurred were determined by the distances at which the observed pattern fell above or below the 95% simulation envelopes, respectively. This analysis does not parse aggregating and overdispering processes; it shows the net resulting pattern.

In addition to analyzing all P. damicornis and S. hystrix colonies, we analyzed P. damicornis <5 cm, ≥5 cm, ≥10 cm, and ≥15 cm in diameter to see if spatial patterns changed with colony size. The <5 cm and ≥5 cm categories were mutually exclusive but because there were not enough colonies between 5 and 10 cm and between 10 and 15 cm to analyze as mutually exclusive groups, larger size categories were subsets of smaller ones.

The 8 × 8 m quadrat surveys could not resolve spatial patterns below the cell size of 0.5 × 0.5 m, meaning that patterns occurring at less than 0.252 m could be undocumented. To determine the spatial distribution of P. damicornis and S. hystrix at smaller scales, we conducted 2 m radius circular surveys around focal P. damicornis and S. hystrix colonies that (i) were the largest colony of that species within 4 m (to reduce the effects of conspecifics), and (ii) occurred where >75% of the substrate within 2 m was suitable habitat for P. damicornis and S. hystrix, again to equalize the likelihood of colonies occurring everywhere in the survey.

The distance to each surrounding (radial) P. damicornis and S. hystrix colony was the average of the distance to that colony’s near and far sides (N = 45 focal colonies for P. damicornis around P. damicornis, 10 for S. hystrix around P. damicornis, and 24 each for P. damicornis and S. hystrix around S. hystrix). We analyzed radial colony counts in 10 cm concentric rings using a generalized linear mixed effects model with Poisson errors and the canonical log link function in R (lme4 package, Bates et al., 2013). Distance was a fixed effect and focal colony was a random effect, with the log10 of the ring sizes as an offset to control for unequal area sampled at each distance (i.e., ring area increased with distance from the focal colony). We repeated this analysis with just the closest 0.5 m and 1 m of the circles in case radial colonies beyond those distances were masking short-range effects of the focal colonies.

We also analyzed the P. damicornis data from the 8 × 8 m plots in the same manner as we did the circular surveys. To convert the plot data, a function was written in R to identify every surveyed P. damicornis colony ≥2 m from all edges of its plot and equal to or larger than a specified diameter (either 15 or 20 cm) as a focal colony (N = 38 and 19 focal colonies, respectively). In order to have an appreciable sample size, we did not restrict focal colonies to those that were the largest within 4 m. The script then calculated the distances to all P. damicornis colonies less than the specified focal colony diameter within 2 m and placed them into 10 cm concentric rings as above. We analyzed the resulting data using the same procedures as described for the circular surveys.

Results

Survival experiments

In our field experiment, neither distance nor direction from focal colony significantly affected survival of P. damicornis or S. hystrix fragments (Figs. 2A and 2B, respectively). We observed two main categories of mortality: bleaching preceding death in place (potentially due to microbes (e.g., Ben-Haim, Zicherman-Keren & Rosenberg, 2003)) and partial or complete disappearance, putatively due to predation (akin to Lenihan et al., 2011). Bleaching (47 and 46 fragments out of 320 for P. damicornis and S. hystrix, respectively) of neither species was affected by distance or direction (Figs. 2C and 2D). Distance and direction did not affect the number of P. damicornis fragments that partially or fully disappeared (putative predation), and direction did not affect this for S. hystrix but distance was significant (z = 2.23, p = 0.03) (Figs. 2E and 2F), with disappearance increasing with distance from the focal colony. In contrast, 0% of the 64 extra fragments of each species remaining on the coral rack where we originally acclimated the corals bleached or disappeared despite being on the same reef at the same time (Cox proportional hazards survival analysis, likelihood ratio for P. damicornis = 16.5, likelihood ratio for S. hystrix = 24.7, p < 0.0001 for both species). Fragments on the coral rack were ∼1 m above the benthos and may have experienced more flow or fewer benthic-associated biotic or physical stressors compared to the fragments on PVC pipes, which were 5–15 cm above the benthos.

Figure 2 Survival and predation of Pocillopora damicornis and Seriatopora hystrix.

(A), (C), (E) are Pocillopora damicornis and (B), (D), (F) are Seriatopora hystrix. Statistical values are from mixed-effects Cox proportional hazards survival analyses. n = 80 fragments at each distance across 10 focal colonies and pooled between both directions. (A, B) Survivorship through time for Pocillopora damicornis and Seriatopora hystrix fragments. (C, D) Cumulative number of fragments that bleached over time. (E, F) Cumulative number of fragments that partially or fully disappeared over time (putative predation).

The rapid disappearance of P. damicornis fragments around some focal colonies suggested spatially localized predation. Therefore, we further divided deaths due to putative predation between isolated predation incidents (disappearance of one or two fragments on a tile in 24 h) and localized predation episodes (disappearance of three or four fragments from a tile in 24 h). We used this classification scheme because three or four fragments tended to disappear from multiple tiles within the same 24 h period at certain replicates, whereas the disappearance of one or two fragments was not often temporally coincident across tiles within a replicate. We distinguished between these two types of putative predation because their causes were potentially different and therefore either one could have been distance-dependent or masked distance-dependence in the other. Six of 10 P. damicornis replicates (23 out of 160 tiles) experienced localized predation on at least one of their eight tiles; three of those experienced localized predation on five or more tiles within 24 h. Two of 10 S. hystrix replicates experienced localized predation (on one tile each). We further investigated localized predation only for P. damicornis because localized predation on S. hystrix was infrequent.

When tiles that had experienced localized predation around three focal colonies were replaced with spare tiles holding healthy fragments, all three sets of replacement tiles again experienced localized predation and their collective survival was significantly lower than that of the replicates that did not experience localized predation in the initial run (mixed effect Cox proportional hazards, z = 3.5, p < 0.0005). Videos of these tiles showed the territorial triggerfish Balistapus undulatus consuming multiple fragments from multiple tiles around two of the three focal colonies. Balistapus undulatus feeding resulted in fragments irregularly broken at or above the top of the epoxy, as was seen for most localized predation episodes in the initial outplanting.

We next examined whether localized predation was distance-dependent and whether it masked distance-dependent mortality in replicates that did not experience localized predation. Distance and direction did not significantly affect mortality in replicates that did not experience localized predation (Fig. 3A). Considering only replicates that experienced localized predation (both original and replacement tiles), neither distance nor direction significantly affected mortality from all causes (Fig. 3B) or just from localized predation (Fig. 3C).

Figure 3 Pocillopora damicornis mortality relating to localized predation episodes.

(A) Survival of Pocillopora damicornis fragments in replicates (4 focal colonies, 32 fragments at each distance) that did not experience localized predation. (B) Survival of P. damicornis fragments in the six focal colony replicates that did experience localized predation and in the replacement replicates. Deaths are from all causes. (C) Fraction of P. damicornis fragments not killed by localized predation episodes in original replicates that experienced localized predation and in the replacement replicates. Direction not shown. Analyses as in Fig. 2.

Pocillopora damicornis fragments were significantly more likely to die of putative predation as opposed to bleach and die in place than were S. hystrix fragments (chi-square test, χ2 = 17.2, df = 1, p < 0.0001). More than three times as many P. damicornis fragments died from putative predation as bleached prior to death (169 vs. 47 out of 320, respectively), while numbers of S. hystrix fragments that died from putative predation versus bleaching did not differ significantly (58 vs. 46 out of 320, respectively). Excluding replicates with localized predation, P. damicornis and S. hystrix appeared equally susceptible to isolated predation and bleaching (χ2 = 0.022, df = 1, p = 0.88).

Distribution surveys

We analyzed patterns of distribution using both entire 8 × 8 m plots and after excluding habitat deemed unsuitable for P. damicornis or S. hystrix (e.g., sand-scoured channels and pools, bommie tops covered in rubble). The analyses using only suitable habitat were quantitatively similar to those using the entire plots but were more conservative. Neighborhood density graphs using only suitable habitat are included here. Neighborhood density analysis indicated that both P. damicornis and S. hystrix were significantly aggregated at up to 1 m when all size classes were considered and surveys from all villages were pooled (Figs. 4A and 4B, respectively). When analyzed by site, the distance below which colonies were aggregated ranged from <1 m in Votua and Vatu-o-lailai to nearly 3 m in Namada. At no distance on any reef were colonies significantly overdispersed.

Figure 4 Neighborhood density analysis of Pocillopora damicornis and Seriatopora hystrix in 8 × 8 m quadrats with replicates from all three reefs combined.

Black lines are observed patterns; grey lines are the 95% simulation envelopes from 999 simulations. Where black lines are above the upper grey line colonies are significantly aggregated, where they are between the grey lines colonies are randomly spaced, and where they are below the lower grey line colonies are significantly overdispersed. These analyses used only areas of suitable substrate (see text for definition).

Identical analyses with P. damicornis separated into size categories (Figs. 4C–4F) indicated that the largest colonies (≥15 cm) were not aggregated at any scale, but all smaller size classes were strongly aggregated at scales of up to 1 m. Thus, smaller colonies appeared to drive the aggregation at up to ∼1 m when we analyzed all sizes together. However, the limited sample size for large colonies (n = 187) may have constrained our ability to detect spatial patterns for large colonies.

To resolve the spatial distribution of P. damicornis and S. hystrix more finely, we conducted separate circular surveys (radius = 2 m) around focal colonies that met specific criteria. Across all 2 m, there was a significant negative relationship between distance from focal P. damicornis colonies and radial P. damicornis count (corrected for area surveyed at each distance and henceforth called density), focal P. damicornis and radial S. hystrix density, and focal S. hystrix and radial P. damicornis density (GLM: z = − 4.4, p < 0.0001; z = − 3.9, p < 0.0005; z = − 3.6, p < 0.0005, respectively) (Figs. 5A and 5B). The relationships within the first 0.5 m or 1 m for these focal-radial combinations were not significant (see Table 1 for all values not provided in text).

Figure 5 Density of Pocillopora damicornis and Seriatopora hystrix at 10 cm intervals around focal colonies.

Density (±SE) of Pocillopora damicornis and Seriatopora hystrix at 10 cm intervals from focal (A) P. damicornis and (B) S. hystrix colonies. The linear regressions shown are to indicate the slope of the relationship found in the generalized linear mixed effects models but do not represent the models’ outputs. Radial colony count significantly declined with distance from focal colony over 2 m for three of the four focal-radial combinations (focal P. damicornis-radial P. damicornis—z = − 4.4, p < 0.001; focal P. damicornis-radial S. hystrix—z = − 3.9, p < 0.001; focal S. hystrix-radial P. damicornis—z = − 3.6, p < 0.001; focal S. hystrix-radial S. hystrix—z = − 1.9, p = 0.06). (C) Density (mean ±SE) of Pocillopora damicornis within 2 m of focal P. damicornis based on the 8 × 8 m surveys. Focal colonies are P. damicornis that are ≥15 cm across or ≥20 cm across. Radial colonies are any colonies below that size. Radial colony count significantly declined with distance over 2 m when colonies ≥15 cm were considered focal (focal colonies ≥15 cm—z = − 3.6, p < 0.0005; focal colonies ≥20 cm—z = − 1.09, p < 0.27).

Table 1 Relationship between radial colony count and distance from focal colony.

Relationship between count of radial Pocillopora damicornis and Seriatopora hystrix colonies and distance from focal P. damicornis and S. hystrix colonies using generalized linear mixed effects models. “Maximum distance” is the distance up to which radial colonies were considered.

Focal species–radial species	Maximum distance	Slope	z value	p-value	
P. damicornis–P. damicornis	0.50 m	−0.0098	−1.0	0.32	
1.0 m	0.00055	0.22	0.83	
2.0 m	−0.0032	−4.4	<0.0001	
P. damicornis–S. hystrix	0.50 m	0.016	0.59	0.56	
1.0 m	0.0012	0.19	0.85	
2.0 m	−0.0057	−3.9	<0.0005	
S. hystrix–P. damicornis	0.50 m	0.0099	0.57	0.57	
1.0 m	0.0042	1.1	0.28	
2.0 m	−0.0036	−3.6	<0.0005	
S. hystrix–S. hystrix	0.50 m	0.042	2.3	<0.05	
1.0 m	−0.00065	−0.18	0.86	
2.0 m	−0.0017	−1.9	0.06	

Across all 2 m, there was no significant relationship between distance from focal S. hystrix colony and radial S. hystrix density (GLM, z = − 1.9, p = 0.06) (Fig. 5B). However, there was a significant positive relationship between distance and density within the first 0.5 m (GLM, z = − 12.99, p < 0.05) but not within the first 1 m.

When we converted the 8 × 8 m surveys into data analogous to the circular surveys and considered any P. damicornis colony ≥15 cm across as a focal colony and any smaller individual as a radial colony, there was a significant negative relationship between distance and radial P. damicornis density (GLM, z = − 3.6, p < 0.0005) across all 2 m but not across the first 0.5 m or 1 m (Fig. 5C and Table 2). However, when the cutoff for focal colonies was 20 cm, there was no relationship between distance and P. damicornis colony count at 0.5 m, 1 m, or 2 m (Fig. 5C and Table 2).

Table 2 8 × 8 survey results converted to radial survey results.

Relationship between count of radial Pocillopora damicornis colonies and distance from focal Pocillopora damicornis colonies using the data from the 8 × 8 m surveys. “Threshold size for focal colony” is the size above which surveyed colonies were designated “focal” and below which colonies were designated “radial.” “Maximum distance” is the distance up to which radial colonies were considered.

Threshold size for focal colony	Maximum distance	Slope	z value	p-value	
15 cm	0.50 m	−0.0064	−0.46	0.65	
1.0 m	−0.0063	−1.5	0.14	
2.0 m	−0.0034	−3.6	<0.0005	
20 cm	0.50 m	0.047	1.2	0.23	
1.0 m	0.00032	0.045	0.96	
2.0 m	−0.0016	−1.09	0.27	

Discussion

Using small portions of adult P. damicornis and S. hystrix colonies to represent ∼6 month old juveniles, we tested for distance-dependent survivorship as a function of proximity to adult conspecifics. Survival experiments with P. damicornis and S. hystrix fragments did not show distance-dependent mortality around conspecific adults (Figs. 2A and 2B). The lack of distance-dependent mortality in this study is consistent with a meta-analysis of distance-dependent mortality studies of the seeds and seedlings of terrestrial plants (Hyatt et al., 2003), in which distance from parents did not affect overall survival. However, when separated by life stage, that meta-analysis found that seedling survival increased with distance from parents while seed survival was not affected, suggesting that the strength of distance-dependent mortality may be a function of age. Our experiment using small coral fragments to represent juveniles attempted to conduct a similar test with corals. Our procedures would not have detected distance-dependent mortality of larvae occurring just after settlement. We would have preferred to conduct a reciprocal transplant experiment of fragments from both corals at differing distances to both conspecific and heterospecific adults but we were unable to gain permission to use that many coral colonies. Thus, we could document spatial patterns of survivorship relative to conspecific adults but not relative to heterospecific adults.

Spatial analyses of the distribution of conspecific colonies might uncover patterns that our short-term experiment could not detect. Observed spatial patterns represent the balance of multiple, potentially opposing processes, such as greater recruit density near brooding parents (similar to terrestrial seed shadows) versus detrimental effects of adult-associated enemies or intraspecific competition on aggregated, nearby recruits. Rather than overdispersion, we found significant clumping within 1 m of conspecifics for both P. damicornis and S. hystrix (Figs. 4A and 4B). The 8 × 8 m surveys and the 2 m radius surveys both supported this pattern; there was a significant negative relationship between P. damicornis radial colony density and distance from focal P. damicornis and a nearly significant negative relationship (with a much more limited sample size) between S. hystrix radial colony density and distance from focal S. hystrix (Figs. 5A and 5B). We also observed a significant negative relationship between S. hystrix density and distance from P. damicornis and P. damicornis density and distance from S. hystrix (Figs. 5A and 5B), suggesting that the cause of declining density with distance need not be species-specific. Since P. damicornis and S. hystrix are confamilial, it is possible that they aggregate because a location that is physiologically beneficial for one might also be beneficial for the other. We did detect one pattern consistent with the Janzen–Connell hypothesis: small colonies of P. damicornis were aggregated at scales of up to 1 m, while colonies ≥15 cm in diameter were not aggregated at any scale (Fig. 4). This selective loss of small colonies near adults is consistent with the Janzen–Connell hypothesis, but is also consistent with self thinning from intraspecific competition without mortality due to enemies aggregated near adults (Zhu et al., 2013).

There are a few potential causes for the observed clumping of conspecifics that could counteract Janzen–Connell effects (Carlon & Olson, 1993). Aggregated settlement near maternal adults may occur for P. damicornis and S. hystrix because brooded planulae can settle quickly after release (Richmond, 1987; Isomura & Nishihira, 2001; Underwood et al., 2007; Torda et al., 2013), and even if planulae disperse meters or kilometers, they may still aggregate near conspecific adults (Babcock, 1988; Tioho, Tokeshi & Nojima, 2001; Doropoulos et al., 2015). Moreover, pocilloporid recruitment is inherently spatially heterogeneous (Dunstan & Johnson, 1998) and occurs in hotspots that may be partially determined by water flow, density of adult confamilials (Eagle, 2006), and substrate suitability (Harriott, 1983; Lee, Walford & Goh, 2009). Thus, multiple ecologically important processes and interactions can generate aggregation of juveniles, and some of these could overwhelm Janzen–Connell effects and make them seem unimportant in the field (at least in the short term), even if they were occurring.

The only other direct test of the Janzen–Connell hypothesis in corals was conducted on planulae and recruits of broadcasting Orbicella (formerly Montastraea) faveolata in the Caribbean (Marhaver et al., 2013). In that study distance-dependent mortality appeared to be microbially mediated, with effects differing upstream and downstream of focal O. faveolata. The design of that study and ours differs in several ways.

First, Marhaver et al. (2013) used planulae and recruits a few days old in their distance-dependent survival experiments, whereas we used fragments taken from mature colonies. There are potential differences between fragments from adult corals and recruits. For example, the physiology, skeletal structure, and microbiomes of fragments from adults may differ from those of recruits and similarly sized juveniles (Vandermeulen & Watabe, 1973; Harriott, 1983; Le Tissier, 1988; Christiansen et al., 2009). The planulae of P. damicornis have higher lipid percentages in their tissue than do adult colonies (Figueiredo et al., 2012), which could affect palatability and microbial defense. Most recruit mortality in the study by Marhaver et al. (2013) appeared to be microbe-related as opposed to predator-generated. In contrast, predators generated considerable mortality of our juvenile sized transplants. Liberally assuming that every bleaching death in our study was due to microbes, only about one quarter of P. damicornis and half of S. hystrix fragments could have died directly because of microbes; thus, about 50–75% of the mortality we observed appeared to be due to consumption by fish. One way in which using colony fragments may not have been so different from using recruits was that both age categories may have similar zooxanthellae endosymbionts, since P. damicornis vertically transfers zooxanthellae to its young (LaJeunesse et al., 2004).

Second, Marhaver et al. (2013) studied the broadcast spawning species O. faveolata, while we studied two brooding species whose planulae may be more likely to settle near their parents, and whose larvae may receive critical components of their microbiome via vertical transmission from adults. Data on the make-up and function of juvenile coral microbiomes are limited but at present there is some evidence that larvae from brooding species may be more consistently endowed with parental components of the microbiome than are the larvae of broadcast spawners (Littman, Willis & Bourne, 2009; Apprill et al., 2012; Sharp, Distel & Paul, 2012; Lema, Willis & Bourne, 2012). In some acroporid corals, juveniles do not develop microbiomes typical of adult colonies until greater than 9 months of age (Littman, Willis & Bourne, 2009) but a core component of the microbiome appears in all the early stages, despite additions of other microbial species from the environment later in development (Lema, Willis & Bourne, 2012). However, in brooding species such as Porites and Pocillopora, critical microbes are transmitted from adults to larvae, or very quickly acquired from the environment, and even very young juvenile stages resemble adults in their composition of key microbes comprising the symbiotic microbiome (Apprill et al., 2012; Sharp, Distel & Paul, 2012). We do not know these relationships for the species we investigated but if their microbiomes take months to develop and are important defenses against microbial enemies, then our use of small adult portions may not mimic juvenile susceptibility to adult-associated pathogens. In contrast, if the critical components of the microbiome are present in even the earliest stages, then our adult fragments should be more representative. We would have preferred to use recently recruited larvae but neither P. damicornis nor S. hystrix planulated at our study site during our experiment.

Finally, the focal adult colonies of Orbicella investigated by Marhaver et al. (2013) form larger, longer-lived colonies than the colonies of Pocillopora and Seriatopora that we investigated. It is possible that larger, longer-lived colonies accumulate more species-specific enemies over their lifetimes; if so, this could more strongly suppress juvenile survivorship near these longer-lived adults.

Although we did not detect distance-dependent mortality, we did document spatially heterogeneous corallivory on P. damicornis. This may promote species coexistence by producing a mosaic of favorable and unfavorable patches for P. damicornis across the reef (Levin & Paine, 1974; Holt, 1984). Corallivore activity can structure coral distribution on reefs in both the Pacific and Caribbean (Neudecker, 1979; Littler, Taylor & Littler, 1989) and parrotfish and butterflyfish density can impact coral recruit and juvenile mortality, respectively (Penin et al., 2010). Localized predation by the triggerfish Balistapus undulatus on small P. damicornis could have a similar effect here. Balistapus undulatus is a generalist with territories of 100–200 m2 (McClanahan, 2000) and eats the tips of branching corals, including P. damicornis (Hiatt & Strasburg, 1960; Neudecker, 1979). This triggerfish species’ territoriality may delineate certain patches on reefs in which some species (e.g., P. damicornis) have high mortality while other species (e.g., S. hystrix) are not directly affected, akin to what is seen with seaweed in territories of the steephead parrotfish on the Great Barrier Reef (Welsh & Bellwood, 2012) or Pocillopora and Pavona in the interaction between damselfish territories and roving corallivores in the Eastern Pacific (Wellington, 1982). Additional experiments are necessary to determine how patchy corallivory contributes to the coexistence of P. damicornis, S. hystrix, and corals in general.

Overall we found little evidence for distance-dependent mortality relative to focal conspecific adults and for the pattern of over-dispersion that distance-dependent mortality would be expected to produce. Instead, both P. damicornis and S. hystrix aggregated at the scale of 1 m or less, with a tendency for small colonies to be clumped around larger ones. These findings suggest that local dispersal shadows or areas of physiological benefit near prospering adult conspecifics equal or exceed Janzen–Connell effects for the brooding corals we studied on these Fijian reef flats. Our experiments using small coral fragments did not detect distance-dependent mortality by species-specific enemies; we did, however, observe spatially heterogeneous corallivory on P. damicornis, which could facilitate species coexistence by delineating reef patches that are more or less favorable to different corals.

We thank the Fijian government and the Korolevu-i-wai district environmental committee for permitting this research, Victor Bonito for local support, and Kristen Marhaver and two anonymous reviewers for helpful comments.

Additional Information and Declarations

Competing Interests

Author Contributions

Field Study Permissions

Data Availability

Mark E. Hay is an Academic Editor for PeerJ.

David A. Gibbs conceived and designed the experiments, performed the experiments, analyzed the data, wrote the paper, prepared figures and/or tables, reviewed drafts of the paper.

Mark E. Hay conceived and designed the experiments, contributed reagents/materials/analysis tools, wrote the paper, reviewed drafts of the paper.

The following information was supplied relating to field study approvals (i.e., approving body and any reference numbers):

Approval for our studies was granted by the Fijian Ministry of Education, National Heritage, Culture, and Arts, Youth, and Sports, and by the Korolevu-i-wai district environmental committee.

The following information was supplied regarding data availability:

Data in BCO-DMO database

Project: Killer Seaweeds: Allelopathy against Fijian Corals

http://www.bco-dmo.org/project/480717.

For individual datasets:

http://www.bco-dmo.org/dataset/564411

http://www.bco-dmo.org/dataset/564397

http://www.bco-dmo.org/dataset/564389

http://www.bco-dmo.org/dataset/564404

http://www.bco-dmo.org/dataset/564418.

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
