# Peer review of "Spatial patterns of coral survivorship: impacts of adult proximity versus other drivers of localized mortality"

_PeerJ, doi:10.7717/peerj.1440_

## Round 0.1 · original submission · Minor Revisions

Both reviewers had a positive response to this manuscript, finding the study a well designed and valuable contribution to the literature, in a well written manuscript. They suggested a few additions or clarifications to the text, some possible considerations of additional analyses and the need to discuss the implications of using fragments as the test subjects. I agree with the positive assessment and will be pleased to recommend publication following minor revisions.

Reviewer 2 provided the majority of his suggestions as inserted comments on a pdf of the manuscript (comments labeled 'system user'). For ease of revision, I provided my own comments on the same version ('DLK' - supplied here as an additional downloadable file for you) to facilitate the revision. In your response to reviewers, please include any substantive comments on the manuscript as well as the reviews. Minor changes in wording do not need to be mentioned unless you disagree and are not following the proposed suggestions.

Reviewer 1 ·

Basic reporting

This study addresses the spatial distribution of corals in a reef flat habitat in Fiji, and specifically explores how spatial location of small corals can affect their mortality. The work is a timely consideration of the Janzen-Connell effect on a tropical coral reef, which was pushed into a brighter light by the important work of Marhaver et al. who studied this effect in the Caribbean coral Orbicella. In short, their study found that coral recruits placed close to adults suffered higher mortality, presumably because of transferred microbes.

The work by Marhaver et al. has made an important contribution to reef ecology, and it is important that tests of similar effects are conducted elsewhere. The present study conducts such work, and finds no clear signs of the anticipated effects for two brooding corals. The study is elegant, well conceived, conducted with precision, and written up in a nice format. I am happy to recommend publication. However, I recommend attention to the following issues:

1. Flow speed would seem to be an important covariate in mediating the J-C effect, potentially by altering the ease by which microbes might hop from one coral to another. Yet there is no mention of flow speeds in the study habitat. I am not familiar with Fiji, but on other Pacific reef flats the flow can be ripping, which would likely dissipate any microbes very quickly.

2. The authors don’t seem to give suitable credit to the limitations of working with tiny coral fragments cut from adult colonies as a proxy for recruits that would be a few days—weeks-months in age. The authors took a very reasonable approach to their study, and their results are meaningful and important. However, it is likely that “genuine” recruits could give different results for many reasons – different symbionts, different tissue age, less accumulated environmental effects, etc. It would be healthy to give a more realistic summary of these limitations.

3. Figs 1, 2 and 3 short change the results and need closer attention to quality and content. They look as if they were quickly cut from the software than generated them, and then dropped into a composite panel. I would recommend cleaning these up, and paying closer attention to what will be visible in print.

Experimental design

No problems here.

Validity of the findings

No problems here.

Additional comments

See above for suggestions for slight improvements. Nice work!

·

Basic reporting

The article appears to comply to the standards of PeerJ. The writing is clear and theory described. Generally, appropriate referencing has been conducted. I do have a few concerns regarding recruits versus juveniles, and a few of the citations. I also think the problem needs to be highlighted prior to the statement of aim. All comments are on the attached pdf.

Experimental design

Generally robust. I have a few queries as to a few of the analyses. Additionally, some of the analyses in the Results are not described in the M&M - or if they were are not clearly presented. All comments are on the attached pdf.

Validity of the findings

Very good.

Additional comments

In general an interesting study of something that has hardly been tested in coral reefs, and the use of these two common brooders a great way to appraoch it. Please find all comments and concerns on the attached pdf and address prior to resubmission.

---

## Round 0.2 · accepted · Accept

The manuscript is now ready for publication. I apologize for the delay. Several manuscripts arrived at a time when I was already very busy.